# When Sugar Reaches the Liver: Phenotypes of Patients with Diabetes and NAFLD

**DOI:** 10.3390/jcm11123286

**Published:** 2022-06-08

**Authors:** Alba Rojano-Toimil, Jesús Rivera-Esteban, Ramiro Manzano-Nuñez, Juan Bañares, David Martinez Selva, Pablo Gabriel-Medina, Roser Ferrer, Juan M Pericàs, Andreea Ciudin

**Affiliations:** 1Endocrinology Department, Vall d’Hebron University Hospital, 08035 Barcelona, Spain; arojano@vhebron.net; 2Vall d’Hebron Institut de Recerca (VHIR), 08035 Barcelona, Spain; jrivera@vhebron.net (J.R.-E.); ramiro.manzano@correounivalle.edu.co (R.M.-N.); jbanares@vhebron.net (J.B.); david.martinez.selva@vhir.org (D.M.S.); 3Medicine Department Bellaterra, Universitat Autònoma de Barcelona, 08193 Barcelona, Spain; 4Liver Unit, Vall d’Hebron University Hospital, 08035 Barcelona, Spain; 5Spanish Network of Biomedical Research Centers, Diabetes and Metabolic Associated Disorders (CIBERdem), 28029 Madrid, Spain; 6Biochemistry Department, Vall d’Hebron University Hospital, 08035 Barcelona, Spain; pgabriel@vhebron.net (P.G.-M.); roferrer@vhebron.net (R.F.); 7Biochemistry and Molecular Biology Department, Universitat Autònoma de Barcelona (UAB), Bellaterra, 08193 Barcelona, Spain; 8Spanish Network of Biomedical Research Centers, Liver and Digestive Diseases (CIBERehd), 28801 Madrid, Spain

**Keywords:** type 2 diabetes, type 1 diabetes, ketone-prone diabetes, MODY diabetes, NAFLD, NASH, liver fibrosis

## Abstract

Type 2 diabetes mellitus (T2DM) and non-alcoholic fatty liver disease (NAFLD) have been traditionally linked to one another. Recent studies suggest that NAFLD may be increasingly common in other types of diabetes such as type 1 diabetes (T1DM) and less frequently ketone-prone and Maturity-onset Diabetes of the Young (MODY) diabetes. In this review, we address the relationship between hyperglycemia and insulin resistance and the onset and progression of NAFLD. In addition, despite the high rate of patients with T2DM and other diabetes phenotypes that can alter liver metabolism and consequently develop steatosis, fibrosis, and cirrhosis, NALFD screening is not still implemented in the daily care routine. Incorporating a clinical algorithm created around a simple, non-invasive, cost-effective model would identify high-risk patients. The principle behind managing these patients is to improve insulin resistance and hyperglycemia states with lifestyle changes, weight loss, and new drug therapies.

## 1. Introduction

Non-alcoholic fatty liver disease (NAFLD) has become the most common chronic liver disease in Western countries. Its incidence is expected to keep growing, parallel to the incidence of metabolic syndrome (MetS) and its determinants. It is estimated that NAFLD affects 25–35% of adults in the general population, and its prevalence increases to 60–80% in obese and diabetic patients [1]. The spectrum of the disease ranges from mild hepatic steatosis and liver inflammation (non-alcoholic steatohepatitis or NASH) to liver fibrosis and cirrhosis that lead to liver-related events and increased overall mortality [2]. 

Several studies demonstrate that insulin resistance (IR) plays a critical role in NAFLD pathophysiology and natural history of the disease [3]. The accelerated lipolysis associated with IR increases hepatic glucose production in NAFLD patients, which upregulates de novo fat synthesis, accelerating NAFLD progression [4]. Other pathogenic pathways such as alterations in lipid metabolism, mitochondrial leakage, and pro-inflammatory cascades’ activation are described in disease progression [5]. Although several drugs have demonstrated efficacy in NASH improvement in early-stage trials [6,7], at present, there are no US Food and Drug Administration (FDA) or European Medicines Agency (EMA) approved drugs for NAFLD.

Within the MetS spectrum, the bulk of research addresses the relationship between either type 2 diabetes (T2DM) or obesity with NAFLD [8,9,10]. However, a growing body of evidence shows that NAFLD is also prevalent in a variety of other forms of diabetes that typically have an earlier onset, such as T1DM, MODY and ketosis-prone diabetes. The lack of clinical awareness of the hepatic disease in these populations added to the scarce research in this area has resulted in the absence of protocols for screening and underdiagnosis of hepatic disease.

This review describes the relationship between DM and non-alcoholic fatty liver disease, the pathophysiological rationale behind this relationship, and its clinical relevance and outcomes. Figure 1 reflects the main mechanisms involved in the physiopathology of NAFLD in different types of diabetes.

We furthermore propose a diagnostic algorithm to optimize diagnosis and contemporize treatment.

## 2. Methodology

A literature review was performed, including the information obtained from the search in the English language of 263 studies published in Ovid MEDLINE, PubMed, EMBASE, and Cochrane database. We defined as the primary outcome to compare between the different DM phenotypes related with NAFLD and the management and screening approaches. The search terms entered were ‘T2DM’, ‘Obesity’, ‘ketone-prone diabetes’, ‘atypical diabetes’, ‘MODY’, ‘mature-onset of the young diabetes’, ‘type 2 diabetes’, ‘diabetes mellitus’ ‘T1DM’, ‘NAFLD’, ‘non-alcoholic fatty liver disease’, ‘NASH’, ‘non-alcoholic steatohepatitis’, and ‘fatty liver’. Included studies were RCT, cross-sectional, longitudinal, or descriptive studies published in peer-reviewed journals between 1987 and January 2022. Inclusion criteria were as follows: published in English, peer-reviewed, addressed NAFLD and T2DM, T1DM, MODY, and ketone-prone DM. Exclusion criteria included abstract or non-peer reviewed articles or studies that were not in the English language or not related to the topic. A total of 197 articles out of 263 were included in the present review after adjusting by inclusion/exclusion criteria.

## 3. Diabetes Clinical Phenotypes

### 3.1. Type 2 Diabetes and Obesity—The Metabolic Syndrome Paradigm

#### 3.1.1. Epidemiology

Type 2 diabetes is the most common form of diabetes. It has an estimated prevalence of 26.9 million people of all ages in the US (or 8.2% of the US population) and 536.6 million people worldwide [11,12]. In Spain, the incidence of DM obtained from the di@bet.es study cohort was estimated at 11.6 cases/1000 person-year (OR (95% CI) 11.1–12.1) [13]. T2DM is characterized by hyperglycemia, IR, and relative insulin deficiency [14]. The primary risk factors associated are those related to lifestyle behaviors such as poor physical inactivity and dietary habits, cigarette smoking, and alcohol consumption [15]. Moreover, overweightness and obesity contribute to approximately 89% of cases of T2DM [11].

T2DM has become an epidemic issue in occidental countries. Due to its asymptomatic onset, it is frequently diagnosed at later stages when micro and macrovascular complications are established [16]. Optimization of management of T2DM, screening in primary care, and advances in research have allowed a decrease in the associated complications [17]. In addition, obesity is directly linked to NAFLD due to IR and metabolic syndrome [18,19].

The 2020 Center for Disease Control and Prevention (CDC) report revealed 34.1 million patients diagnosed with diabetes among adults aged 18 years or older in the United States, with type 2 diabetes accounting for 90% to 95% of cases. Among the reported cases, 45.8% of individuals had obesity (body mass index (BMI) of 30.0 to 39.9 kg/m^2^), and 15.5% presented extreme obesity (BMI of 40.0 kg/m^2^ or higher) [11].

In global studies, the prevalence of T2DM is estimated at 10.5% [12], which has steadily increased during the last two decades due to changes in nutrition and physical exercise in Western countries. According to current predictions, the incidence of T2DM is expected to rise from 171 million in 2000 to 366 million individuals by 2030 [20]. One of the main drivers of such worrying forecasting is the increased prevalence of childhood obesity, with approximately 35.1% and 26.5% of children being overweight or obese in the US, respectively [21]. Social determinants of health such as gender, race/ethnicity, and socio-economic status expose significant disparities amongst T2DM patients. For example, Black and Hispanic children have higher risk-adjusted obesity odds when compared to Asians or Caucasians [22]. Similar results are described for T2DM, with the highest prevalence among people of Hispanic origin (12.5%) and Blacks (11.7%) [11]. Low socioeconomic status has been found also associated with poorer outcomes [23]. Lifestyle behaviors, e.g., physical inactivity, smoking, or alcohol consumption, are also strong drivers of T2DM, independently of genetic predisposition [15,24].

#### 3.1.2. Pathophysiology

Type 2 diabetes mellitus (T2DM) and non-alcoholic fatty liver disease (NAFLD) have been traditionally linked to one another. The main pathophysiological link between T2DM and NAFLD is IR [25], and hepatic steatosis and fibrosis are also related to the development of IR, which substantially increases the risk of subsequent T2DM [3]. On one hand, hyperglycemia was demonstrated to be a hazard in hepatic steatosis and fibrosis development: several in vitro models demonstrate the role of hyperglycemia in the process, e.g., Robin C et al. in 2017, demonstrated that hyperglycemic cell culturing conditions induced steatosis within a human hepatocyte cell line (HepG2 cells) by the accumulation of intracellular lipids [26]. Experimental models, such as in streptozotocin-induced diabetic models in mice, show that progression in NAFLD is due to hyperglycemia-induced inflammation [27]. An increase in glucose substrate in the hepatocyte promotes the accumulation of free fatty acids in the cell and the stimulation of hepatic lipogenesis by upregulation of the Krebs cycle and an increase in the expression of ChREBPnad liver X receptor alfa. This cascade activates downstream fibrogenic pathways with oxidant stress and inflammasome activation that leads to cell apoptosis by cytokines such as IL-1B, IL-6, or TNF-alfa that finally result in inflammation, hepatocyte injury, and liver fibrosis [28,29,30]. On the other hand, the excess of triglyceride synthesis is another pathophysiological landmark of NAFLD. IR was proven to be responsible for an increase in adiposity in hepatic cells, as demonstrated in studies with the hyperinsulinemic-euglycemic clamp [31] and indirectly measured through the HOMA-IR index [32]. These changes result in a boost of gluconeogenesis cascades, an increase in free fatty acid (FFA) levels, and simultaneously impairment in hepatic glycogen synthesis. These metabolic changes lead to liver fat accumulation, resulting in a preferential shift from carbohydrate to FFA beta-oxidation [33]. FFA accumulation and IR activate several inflammatory pathways related to pro-inflammatory cytokines such as TNFα or IL-6 [34]. Other extrahepatic drivers of liver inflammation include IL-1B, or IL-6 liberated from adipose tissue in obese individuals [35]. These cytokines create a pro-inflammatory environment producing hepatotoxic free oxygen radical species that result in hepatocellular injury and fibrosis [36].

Despite of that, not all patients with NASH exhibit IR. This suggests that other factors may influence the progression to NASH. For example, genetic variants are strongly associated with histologic severity. Petersen et al. demonstrated that specific polymorphisms in the gene encoding for apolipoprotein C3 are markers of hepatic IR and inflammation [37], such as other research related IL-6 polymorphisms [38]. Patatin-like phospholipase domain-containing protein 3 (PNPLA3) is a lipid droplet-associated protein that is increased in obese patients. Higher levels are associated with augmented liver fat content [39]. Finally, hepatic lipid metabolism disruptions, inflammation in adipose tissue, and ectopic sites of fat deposition interfere with normal hepatic function and give way to excess storage of lipid droplets in hepatocytes [40]. In fact, de novo lipogenesis is threefold higher in patients with NAFLD [41]. In addition, hepatocellular FFA accumulation is sustained by impaired synthesis and secretion of very-low-density lipoproteins and excessive importation of FFA from adipose tissue [42]. The activation of these cascades eventually leads to impaired antioxidant capacity with increased oxidative stress and mitochondrial function defects, leading to hepatic fibrosis and IR [36].

#### 3.1.3. Clinical Manifestations

T2DM and obesity are among the most important risk factors of NASH, liver-related events, hepatocellular carcinoma (HCC), and mortality in patients with NAFLD [43,44,45,46]. The prevalence of NAFLD in T2DM patients is over 50%, and when both T2DM and obesity coexist, the prevalence of NAFLD ranges between 60% and 80% [47,48,49,50]. A global meta-analysis showed that T2DM was present in 22.51% of patients with radiologically defined NAFLD and 43.36% among patients with NASH determined by biopsy. Of note, 56% of the individuals with histologically proven NASH had normal liver enzymes [2,43]. Data recollected from the Edinburgh Type 2 Diabetes Study (ET2DS) registered a prevalence of 42.6% of NAFLD in patients with T2DM. Independent predictors were BMI, duration of diabetes, HbA1c, triglycerides, and metformin use [51]. These studies also showed that some factors play a critical role in the relationship between NAFLD and diabetes. For example, compared to T2DM only, patients with T2DM and NAFLD were more likely to present hypertension, dyslipidemia, and cardiovascular diseases [52] which means that the presence of NAFLD determines a higher risk of poor cardiovascular and metabolic outcomes, a fact that was proven in previous studies. Indeed, a meta-analysis including 16 studies representing more than 34,000 patients demonstrated that NAFLD was associated with a higher cardiovascular risk and conferred a more elevated risk adjusted-odds of cardiovascular events, including myocardial infarction and stroke [53].

Emerging data suggest that up to 20% of patients diagnosed with NASH will develop cirrhosis in the disease course [54]. The fibrosis stage is the main predictor of liver-related illness, liver transplantation, and liver-related mortality [2,43]. Both T2DM and overweight/obesity are predictors of advanced fibrosis and comorbidities in patients with NASH [45,55] with a >10-fold risk of HCC [56]. Remarkably, and contrary to other causes of chronic liver disease, HCC can develop in non-cirrhotic livers in the setting of NAFLD. In addition, patients with obesity have an increased incidence of HCC without cirrhosis compared with non-obese patients [57,58]. Numerous studies linked pro-inflammatory cytokines derived from IR and hyperinsulinemia to activation of carcinogenic pathways [59]. Yet, cardiovascular events such as stroke or myocardial infarction are the leading cause of death in patients with NAFLD [60]. NAFLD also is shown to worsen the prognosis of T2DM, e.g., to worsen glycemic control and microvascular complications such as nephropathy and retinopathy [61,62]. To remark, another factor with a close relationship with metabolic syndrome is the presence of hepatitis virus C infection (HCV). Recent evidence from prospective studies suggests that the treatment of HCV in NAFLD patients with and without fibrosis, would reduce the risk of T2DM by 81% by restoring normal glucose metabolism. The same effect was reported for cardiovascular events such as acute coronary syndrome, myocardial infarction, stroke, or transient ischemic attack, with a decrease between 56 and 75 cases per year for 10,000 HCV patients with long-term remission, independently of other risk factors such as smoking, dyslipidemia, or hypertension [63,64,65].

#### 3.1.4. Management: Diagnosis and Interventions

All patients with NAFLD should be screened for T2DM, and anthropometric data must be recorded. Most T2DM patients are asymptomatic; thereby, early screening will help these patients prevent complications. The American Diabetes Association (ADA) criteria for the diagnosis of T2DM focus on the following tests [66]:-Glycated hemoglobin (HbA1c) measures the average blood sugar levels over the prior 90 days. Hb1Ac 6.5% or higher is diagnostic of T2DM [67].-A fasting plasma glucose of more than 126 mg/dL is a very specific parameter, but its sensitivity has been reported to be unsatisfactory due to false negatives among individuals with impaired glucose tolerance [68].-The oral glucose tolerance test measures the blood sugar level before, and two hours after, 100 g glucose overload; the diagnosis is established when blood sugar is equal to or greater than 200 mg/dL [69].

In patients with symptoms suggestive of diabetes, a random plasma glucose test equal or greater to 200 mg/dL is sufficient for T2DM diagnosis. A three-year control at age 45 should be carried out according to the ADA, especially in people who are overweight or obese. These screenings should be conducted annually if risk factors are present: family background of DM, prediabetes, personal history of gestational diabetes, dyslipidemia, or age greater than 65 years old [66].

Although the concomitant occurrence of T2DM and hepatic disease is becoming increasingly common, both the ADA and European Association of Diabetes (EASD) recommendations for screening NALFD are still not implemented in the daily care routine. Incorporating a simple clinical diagnostic algorithm formed by a simple, non-invasive, and cost-effective test would allow for identifying patients who need a liver biopsy. Relying on a cohort of 1249 patients with histology-proven NASH, Bazick et al. developed a score to identify T2DM patients at risk of NASH, showing a specificity of 90.0% and a sensitivity of 56.8%. This score includes BMI, waist circumference, HbA1c, HOMA-IR, and serum levels of transaminases, albumin, and ferritin [70]. Other scoring systems implemented for the general population, such as FIB-4, with a sensitivity of 69% and specificity of 77% [71] and NAFLD fibrosis score [72], were demonstrated to be valuable tools for evaluating the risk of hepatic fibrosis [73,74] as first-line non-invasive tests and support decision-making regarding referral to a liver specialist and performing a second-line test such as transient elastography.

Abdominal ultrasonography is the first-line imaging technique recommended to identify steatosis. It is considered the most cost-effective non-invasive tool for large-scale fatty liver screening [75,76,77]. Even though non-invasive methods are currently available, the definite diagnosis of NAFLD/NASH still relies on liver biopsy and histologic examination [78].

Treatment should not be deferred in patients with NAFLD, especially if significant fibrosis is present. Some general measures for these patients include abstinence from alcohol [79], vaccination for hepatitis virus A and B [80], and modification of risk factors for cardiovascular disease [81]. Several studies examined the role of weight loss on NAFLD progression. A weight loss of at least 5 percent of body weight demonstrated an improvement in hepatic steatosis, and a weight loss ≥7% also improved fibrosis and inflammation [82,83]. Another prospective study recruited 293 patients with histologically proven NASH and were endorsed in a one-year program with restriction calorie intake to reduce their weight; although only 30% of patients achieved the goal, a 5% weight loss showed histological improvement, and patients who lost ≥10% of their weight had a more dramatic regression in fibrosis [84]. The first line therapy for patients with T2DM and obesity includes an integrated diet and exercise plan. A randomized, cross-over 6-week dietary intervention study of patients with biopsy-proven NAFLD showed that the Mediterranean diet induced a reduction in liver steatosis and insulin resistance, with subsequent beneficial effects on inflammation and fibrosis [85]. Additionally, physical activity is linked to better outcomes in NAFLD. A longitudinal study made by the US National Health and Nutrition Examination Survey from 2003–2006 to 2015 demonstrated that exercise was associated with lower risk of all-cause mortality and lower risk of cardio-vascular-disease-related mortality [86]. For patients with NASH who do not meet their weight loss goals, additional treatments such as bariatric surgery are proposed. At present, there is a lack of randomized clinical trials that demonstrate the effectiveness of bariatric surgery in NAFLD patients. Although some studies showed improvement in inflammation and fibrosis following bariatric surgery in patients with NASH, the methodology is heterogeneous [87,88] Furthermore, recent studies showed worsening of NAFLD/NASH features in some patients following bariatric surgery. Because of that, it is mandatory to monitor liver enzymes and semiology in all patients postoperatively [89].

Some antidiabetic therapies are efficacious in treating NASH. For example, pioglitazone improved histologic features of NASH in biopsy-proven NASH patients. In a meta-analysis of 4 trials that compared thiazolidinediones with placebo in 334 patients with NASH, patients in treatment with thiazolidinediones had a reduction in alanine aminotransferase levels (−10.9 vs. −36.2 u/L; *p* = 0.009), gamma-glutamyltransferase levels (−9.4 vs. −41.2 u/L; *p* = 0.002), hepatocellular injury (*p* = 0.005), and fibrosis (*p* = 0.05) [90]. However, liver specialists seldom prescribe pioglitazone to treat NAFLD/NASH. Glucagon-like peptide-1 (GLP-1) receptor agonism seems to be a promising pharmacological approach, with preliminary data from Phase 2 RCT for both liraglutide and semaglutide [9,10]. Metformine is the antidiabetic agent most commonly used in T2DM patients. Although these biguanides demonstrated an effective reduction in hepatic and peripheral insulin resistance [91], a difference in reduction in hepatic fat content or a decrease in inflammatory markers was not observed in several studies [92,93,94]. Other groups of diabetic agents that demonstrated lack of clinical efficacy are SGLT-2 inhibitors, although few RCT are registered regarding its hepatic effects. At present, only empaglifozin induced a significant reduction in liver fat [95,96], but not a statistical difference in fibrosis compared with control groups. Other new agents are being tested, such as dual PPARα and PPARδ agonist elafibranor (or GFT505) that showed a resolution of steatohepatitis without worsening fibrosis, with a favorable safety profile [97].

### 3.2. Type 1 Diabetes (T1DM)

Although it appears that the most common liver disease in patients with T1DM is NAFLD [98], the links between both disorders remain largely unelucidated. T1DM is an autoimmune disorder that typically presents in children and younger adults. Until recently, there was not a traditional association between obesity, metabolic syndrome, and T1DM [14], yet with the increasing prevalence of obesity in the general population, a parallel rise has been observed in patients with T1DM and overweight or obesity [99]. Moreover, a pathogenic role of obesity in T1DM was recently described, related to β-cell stress, ectopic adipose tissue, and an increase in autoimmune disorders [100,101,102].

#### 3.2.1. Epidemiology

A global increase in T1DM incidence has been observed, with a 2–3% increase per year [103,104,105]. The higher incidence is found among children younger than five years, and variations occur according to environmental and behavioral factors such as diet, obesity, vitamin D sufficiency, gut-microbiome changes, or exposure to certain viruses [100,105]. As in the case of T2DM, socioeconomic factors play a paramount role in the differences in the prevalence among genetically similar patients [106]. Although T1DM can also develop in adulthood, the higher incidence of T2DM among adults and the lack of strong diagnostic criteria make such late diagnosis rare and challenging [107].

#### 3.2.2. Pathophysiology

Although T1DM and T2DM share hyperglycemia as their landmark, various underlying mechanisms primarily differentiate both entities. While an absolute insulin deficiency characterizes T1DM, the natural history of T2DM is marked by IR in peripheral tissues. Yet, T2DM can also lead to insulin deficiency when insulin production is depleted due to exhaustion of pancreatic synthesis [14,108]. As in T2DM, the state of hyperinsulinemia found in T1DM affects glucose and lipid liver metabolism and triggers pro-inflammatory cascades that produce liver fibrosis and cirrhosis [3,29,109,110].

A proposed differential mechanism of liver damage in T1DM relates to the increase in visceral adiposity secondary to frequent snacking in the hypoglycemia context, which increases caloric and fructose intake [111]. Other mechanisms that may explain the development of hepatic complications in patients with T1DM are the relative portal vein insulin deficiency that leads to altered hepatic glycogen storage, absence of inhibition of hepatic gluconeogenesis, and overall metabolic disturbance with a shunt to lipogenic pathways [14,112,113]. The lack of endogenous insulin also alters the dynamic of insulin delivery with more minor variation in the plasma range, ending in downregulation of insulin receptors on hepatic cells due to continuous exposure [114]. This altered pharmacokinetics implies the development of a relative IR with the subsequent hepatic damage [109].

As mentioned above, not all patients with NAFLD are associated with obesity and MetS. This statement acquires relevance in T1DM due to its genetic footprint. Several genetic risk alleles associated with fatty liver disease are described in the literature, including those containing the genes PNPLA3 and transmembrane six superfamily member 2 (TM6SF2) [115]. Other potential genes that are related to T1DM and insulin-dependent T2DM are those that encode sterol regulatory element-binding protein (SREBPs) and carbohydrate-responsive element-binding protein (ChREBP). Alterations in these proteins stimulate lipogenic and glycolytic pathways that contribute to NAFLD [116,117].

#### 3.2.3. Clinical Manifestations

Despite T1DM being at an increased risk of developing NAFLD, likely higher than T2DM patients (both because of the intensity of metabolic dysfunction and the earlier onset of the disease), a limited number of studies are addressing the prevalence of NAFLD in T1DM. A meta-analysis by Vries et al. reported an overall prevalence of 19.3% in T1DM, which increased to 22% in adults [98]. A recent study showed a NAFLD prevalence of 16–21% in T1DM patients by share wave elastography at liver ultrasound [118].

Evidence on the outcomes of patients with NAFLD and T1DM is also preliminary. In a cohort of 4641 patients with T1DM, Harman et al. found that overall, patients with T1DM were at increased risk of developing liver cirrhosis compared to the general population [119]. Targher et al. demonstrated that NAFLD increases the risk of both microvascular and macrovascular complications in T1DM, i.e., chronic kidney disease (37.8% vs. 9.9% in patients without NAFLD), retinopathy (53.2% vs. 19.8%), coronary artery disease (10.8% vs. 1.1%), cerebrovascular (37.3% vs. 5.5%), and peripheral vascular disease (24.5% vs. 2.5%, *p* < 0.001 for all comparisons) [53,120]. Further prospective studies are needed to evaluate whether NAFLD is an independent factor for hepatic and cardiometabolic complications that might contribute to the excess mortality in T1DM cohorts [121,122].

#### 3.2.4. Management: Diagnosis and Interventions

T1DM is characterized by autoimmune β-cell dysfunction and loss compared to patients with T2DM [14]. Typically, these patients present hyperglycemic symptoms at onset as polyuria, polydipsia, and body weight loss. An estimated 26.3% to 31.7% of patients debut with a life-threatening diabetic ketoacidosis due to the total absence of insulin production [123,124]. The definite diagnosis is confirmed through positive autoimmunity (including insulin, glutamate decarboxylase, islet antigen 2, zinc transporter eight, and tetraspanin-7 autoantibodies) [110], yet in around 10% of patients who do not express such antibodies, low C-peptide measurements also confirm the diagnosis [125].

The cornerstone of drug therapy in T1DM is exogenous insulin administered through either subcutaneous injections or an insulin pump. The objective is to mimic physiologic insulin release with a basal dose that controls glycemia overnight and between meals, as well as bolus doses that cover carbohydrate rations when feeding [126]. These patients require close monitoring with dose adjustment for physical activity, illness, or stress. Management of T1DM rapidly changed over recent years, with continuous glucose monitoring, intermittently viewed devices, and closed-loop systems, and an artificial pancreas leading to substantial improvements in glycemic control and quality of life [127,128,129,130].

There is a current increase in obesity prevalence, IR, and MetS among T1DM patients, yet NAFLD screening is not systematically recommended in patients with T1DM in the recent ADA 2022 guidelines [131]. However, it is reasonable to recommend yearly NALFD screening in this group, as proposed for patients with T2DM. In addition, the use of antidiabetic drugs in patients with T1DM and NAFLD is controversial. Previous studies demonstrated that endogenous GLP-1 levels are low in T1DM patients [132]. The levels of GLP-1 correlate with the presence of metabolic syndrome [133,134]. Despite favorable results in trials with liraglutide [135], GLP-1 analogs are not approved in T1DM. In NAFLD cohorts, GLP-1 agonists demonstrated a reduction in liver steatosis and IR [136]. The potential effect on hepatic fibrosis and reduction in liver fat content in this cohort of patients remain unexplored, and the current standard of care for NAFLD in T1DM relies on ensuring optimal glycemic control besides diet/weight loss and physical exercise. Interestingly, HbA1c appears not to be a good predictor of NAFLD development in T1DM [82]; hence, new parameters obtained with glycemic control devices such as time in range and coefficient of glycemic variation that are strongly associated with diabetic complications might be explored [137,138,139].

### 3.3. MODY Diabetes

MODY is a mild, largely asymptomatic form of diabetes that occurs in nonobese children and young adults with a dominant inheritance pattern [140]. This form of diabetes is commonly misdiagnosed as T1DM or T2DM and is often inappropriately managed with insulin, whereas the adequate treatment consists of a sulfonylurea [141,142].

#### 3.3.1. Epidemiology

MODY represents a clinically heterogeneous form of β-cell dysfunction caused by genetic mutations with an autosomal dominant form of inheritance. Because of the diverse patterns of presentation and the need for costly molecular diagnosis, it is frequently misdiagnosed as other types of diabetes [142,143]. HNF-1α/MODY3 and GCK/MODY 2 are the most common mutations [144,145]. One of the first prevalence studies estimated 35.2 cases per million with a confirmed genetic test in the UK [146]. In a study carried out on Norwegian diabetes registers, MODY accounted for 0.4% of patients [147]. In the US, the estimated prevalence is 2.1 per 100,000 individuals younger than 20 years [148]. In these groups, we will include a subtype of inherited diabetes associated with lipodystrophies that are associated with severe insulin resistance, premature diabetes, hypertriglyceridemia, and hepatic steatosis due to defects on adipocyte development, differentiation, and apoptosis [149,150]. Although acquired lipodystrophies related with chronic corticosteroids therapy or human immunodeficiency virus (HIV) infection are more common in the general population [151], this group is characterized by a translocation of subcutaneous adipose to ectopic parts of the body, including the liver, which subsequently drives to hepatic inflammation and fibrosis. Therefore, NAFLD has been described in these patients, as confirmed in histologically confirmed cohorts, with a prevalence around 82–90% [152,153,154]. Clinicians must be aware that some of these patients are characterized by low weight and BMI; therefore, recent guidelines recommend the screening of lipodystrophy in lean individuals with a diagnosis of NASH [155]. The severity of NAFLD will depend on the type of lipodystrophy, being more severe in generalized lipodystrophy, but also on specific mutations, such as LMNA mutations [156].

#### 3.3.2. Pathophysiology

MODY was recognized as a disease in 1964 at the Fifth Congress of the International Diabetes Federation in Toronto, and since then, significant progress has been made in understanding its pathophysiology [157]. In 1974, Tattersall et al. reported a new form of diabetes with an autosomal dominant pattern of inheritance that typically presented in young patients who could discontinue insulin therapy over the course of the disease [158]. Since 1975, various mutated genes were identified and associated with different subtypes of MODY with a wide diversity of clinical features, severity of hyperglycemia, and age onset [159]. GCK (MODY 2) and HNF1A (MODY 3) mutations account for almost 70% of all cases of MODY, followed by HNF4A (MODY 1). HNF1A and HNF4A genes encode the transcription factors hepatocyte nuclear factor-1 alpha and factor-4 alpha [160] and coordinate gene expression of proteins involved in glucose transport and glucose metabolism, and β-cell apoptosis, which lead to an increase in cellular apoptosis and defects in insulin secretion [161]. The Glucokinase gene is responsible for detecting bloodstream glucose by its transformation to glucose-6-phosphate by the glucose transporter 2 (GLUT2). Heterozygous mutations imply less function of this enzyme, and affected β-cells are less sensitive to glucose variations that result in elevated fasting and postprandial blood sugar.

#### 3.3.3. Clinical Manifestations

Patients diagnosed with the most common forms (MODY 1, 2, and 3) have a similar risk for complications as those with T1DM and T2DM. Therefore, an optimal glycemic control is inversely related to poor micro- and macrovascular outcomes [161]. Patients carrying heterozygous mutations rarely develop micro- or macrovascular complications [162]. Concerning the association of MODY and NAFLD, again, the body of evidence is overall poor. Multisystemic forms that imply alterations in the transcription factor hepatocyte nuclear factor 1β (HNF1B) (MODY 5) with clinical features that include early-onset diabetes mellitus, pancreatic hypoplasia, genital tract, kidney hypoplasia, cognitive impairment, and abnormal liver function might be the subgroup at higher risk of NAFLD. Hepatic dysfunction is presented in 65% of patients, and the classical phenotype is characterized by elevated serum transaminases, steatosis, and periportal fibrosis. Yearly abdominal ultrasonography and biannual laboratory monitoring are proposed for patients with MODY 5 [163,164,165].

#### 3.3.4. Management: Diagnosis and Interventions

Molecular diagnosis of MODY should be performed in patients with phenotypical features: young nonobese patients with absence of pancreatic autoimmunity with low or null insulin requirements and strong familiar association [166]. Some experts proposed assessing endogenous C-peptide levels on serum or urine samples in patients with T1DM to identify those who may benefit from MODY genetic testing [167]. GCK-MODY onset typically occurs in pregnant women. These patients usually have an altered fasting glycemia with an increase in oral glucose tolerance (OGTT) less than 4.6 mmol/L [168]. HNF1A and HNF4A MODY present during adolescence or young adulthood, and large glucose increases are observed on OGTT, but fasting blood glucose levels are usually normal. These diabetes subsets present an excellent response to sulfonylureas [169]. Finally, the MODY 5 form has an earlier onset with absolute pancreatic deficiency and the need for insulin therapy. In addition, extra-pancreatic alterations are present such as liver disturbances and genital tract and urinary malformations [170].

### 3.4. Ketone-Prone Diabetes (KPD)

In 1987, Winter et al. reported a new form of diabetes called ‘atypical diabetes’ by then [171]. The discovery led to a rise in recognizing this uncommon clinical presentation, with an abrupt onset—typically with ketoacidosis—and transient insulin requirements that usually occurred in African-American or Hispanic patients and were associated with obesity and a strong family history of type 2 diabetes [172,173,174,175]. Some evidence suggests that a primary glucose desensitization acts as a trigger to β-cell exhaustion and dysfunction that lead to acute metabolic failure. Initial aggressive treatment with insulin therapy and antidiabetic drugs such as metformin predict better outcomes and a delay in the recurrence of hyperglycemia [176,177].

#### 3.4.1. Epidemiology

These patients are often Afro-American and Hispanic, obese, middle-aged men, with a family history of type 2 diabetes. Most prevalence studies on KPD made are US-based, with an estimate of 20% and 50%, respectively, in African American and Hispanic patients with new diagnoses of diabetic ketoacidosis [171,173,178,179]. Obesity is present in 56% of newly diagnosed patients, and more than 80% of patients have a family history of T2DM [180,181]. More recent studies estimated an average incidence of 60% among patients attending emergency rooms due to ketoacidosis [182]. The prevalence of this diabetes seems to be lower in Asian and White Americans, representing less than 10% of cases of diabetic ketoacidosis [183,184]. Patients with KPD present with acute IR, elevated glucose, Hb1Ac, and ketone levels, and unlike T1DM, they do not exhibit β-cell antibodies.

#### 3.4.2. Pathophysiology

The natural history of KPD has two phases. These individuals initially present an acute form presentation with severe hyperglycemia and ketosis due to a lack of response and stimulus of β-cell insulin secretion [178]. The causes of these acute onsets in patients affected with ketosis-prone diabetes remain unknown. Patients present a diminished insulin and c-peptide response to an oral glucose load in the second phase. In studies with a euglycemic insulin clamp, there was no difference in baseline glucagon levels and glucagon suppression between KPD patients and the normal controls. In short term follow-up, the insulin secretion increased after a few weeks of exogenous insulin treatment, and after months, there was no difference in the beta-cell response and the insulin secretion compared to controls [185]. IR is not systematically found and appears to be associated with ethnicity and geographic variability [186,187].

#### 3.4.3. Clinical Manifestations

The typical presentation is a new clinical onset with severe hyperglycemia with high ketones or diabetic ketoacidosis, negative GAD, and islet cell autoantibodies. The type and rate of complications are similar to T2DM [162,164,165,166]. Provided that IR characterizes KPD and related metabolic abnormalities, pathogenic liver pathways that contribute to NAFLD are certainly stimulated; yet, there are no specific studies of NAFLD in KPD. It is worth underscoring that African Americans with T2DM are at lower risk for hepatic steatosis than White Americans, which is not explained by ethnic differences in BMI, HOMA-IR, or toxic or drug ingestion [55,180]. In contrast, Hispanics are at higher risk of steatohepatitis, seemingly due to the higher prevalence of obesity and IR [188,189].

#### 3.4.4. Management: Diagnosis and Interventions

Accurate clinical history and physical examination are essential to distinguish KPD from T2DM. Central obesity and acanthosis nigricans in skin folds related to IR are more prevalent in KPD [190]. The diagnosis of asymptomatic forms could be made with the HbA1c test, OGTT, and two basal altered glycemia as recommended for T2DM [57]. Intensive insulin and fluid replacement may be necessary at onset [165,170]. A significant percentage of patients will need exogenous insulin after several weeks since they can be managed with oral agents. Sulfonylurea agents have been proved efficacious and safe in the longstanding treatment of KPD [180,182]. There is a lack of studies on SGLT2 inhibitors, and because of its direct link with euglycemic ketoacidosis, further data are needed before being recommended to patients with KPD.

## 4. Algorithm of Diagnosis and Treatment

Numerous guidelines such as those of the ADA [131], the European Association for the Study of Diabetes, the European Association for the Study of the Liver [191], or the American Association for the study of Liver Disease [192] recommend yearly screening of NAFLD in diabetic patients. However, there are relevant controversies on fundamental aspects of screening, such as the non-invasive test to be used, whether it should be carried out systematically or on a case-finding basis, or whether the approach might or might not be the same at the primary care level and the diabetes clinic regarding when to refer to the liver specialist, etc. [193,194]. Hence, widespread systematic NAFLD screening is not still implemented in many countries. Although a FIB-4 cut-off <1.3 is considered to rule out advanced fibrosis in the general population accurately, and thus it is recommended by the latest EASL guidelines on the use of non-invasive tests [195], Boursier et al. recently found that a substantial proportion of diabetic patients with NAFLD and FIB-4 <1.3 who underwent liver biopsy had F3-F4 fibrosis [196]. Therefore, in diabetic patients with FIB-4 < 1.3 and concomitant factors that might lead to the suspicion of significant or advanced fibrosis (e.g., age, obesity, poor control of DM, elevated transaminases, or other altered non-invasive serologic tests for fibrosis such as ELF), it might be reasonable to perform liver elastography to assess liver stiffness or directly refer the patient to the hepatologist clinic. At this moment (Figure 2), we propose an algorithm that will indeed be refined in the following years as further studies shed light on critical points of NAFLD screening and referral pathways.

The scope of the proposed algorithm is limited as there is still much uncertainty regarding the most accurate non-invasive method to detect NAFLD. On the other hand, studies show that biopsy is a reliable method to detect and stage the degree of fibrosis once NASH is established and the only one to detect active inflammation or steatohepatitis [197], which is known as a prevalent feature of liver damage among diabetic patients and in some cases the only one. Additionally, a diagnostic approach based on liver biopsy was demonstrated to add a survival benefit compared to a non-invasive-based approach. However, liver biopsy is not without perils, and we believe that biopsies should be reserved for patients with higher values of liver stiffness on transient elastography. The decision to perform a liver biopsy should be individualized and taken by an expert hepatologist.

Currently, there are no NAFLD-specific pharmacologic therapies approved for widespread use, and the ones that do show beneficial effects are available in randomized clinical trials. Some of these drugs were tested in diabetic patients and are commercially available. For example, GLP-1 agonists, a part of the routinely pharmacological armamentarium for diabetic patients, show promising results in phase-2 randomized clinical trials. In a randomized clinical trial of semaglutide in patients with non-alcoholic steatohepatitis, Newsome et al. [7], showed that once-daily semaglutide resulted in a higher proportion of patients with NASH resolution without worsening of fibrosis and higher loss of body weight after 72 weeks of treatment. Liraglutide demonstrated similar properties in a randomized clinical trial where 39% of patients treated with the drug had NASH resolution compared to 9% of patients treated with a placebo [6].

## 5. Concluding Remarks

In recent years, in parallel to the increase of its prevalence worldwide, the knowledge of NAFLD diagnosis, pathophysiology, and treatment has grown exponentially. The relevance of NAFLD as a global public health threat is reinforced by its parallel evolution alongside DM, obesity, and MetS. This review aimed to underscore that different DM phenotypes have various types of liver involvement that need personalized management. Despite the lack of approved drugs by the FDA and EMA to treat NAFLD, several pharmacological treatments are used to treat DM, obesity, or dyslipidemia that are shown to be efficacious in treating NAFLD; yet, the evidence is still preliminary. Unfortunately, in the diabetes field, there is no consensus on how to screen for NAFLD. In the hepatology field, there are gaps in the recommendations for liver fibrosis screening in diabetic patients. The gold-standard method to evaluate the presence of NASH is the liver biopsy, as we could observe in some prospective studies that demonstrated a prevalence of 58.52% among patients with metabolic syndrome and 96.82% in T2DM patients [197]; other studies recorded lower prevalence such as that registered in a global systematic review performed in 2017 with a value of 54% [44]. Transient elastography is the screening test more implemented in the daily care routine, although it has a moderate sensitivity compared with the results of biopsy. In a meta-analysis published in 2013, this test reported a high diagnostic accuracy, with an Area under the curve (AUC) in the range of 0.84–0.87 for fibrosis stage ≥ F2, 0.89–0.91 for fibrosis stage ≥ F3, and 0.93–0.96 for fibrosis stage F4 [197], which was also recommended for NAFLD because of its cost-effectiveness and the lack of adverse events. We proposed a simple algorithm for early intervention in patients with DM and suspected NASH. Further studies are needed to improve referral pathways, work-up, and the standard of care for diabetic patients with NAFLD.

## Figures and Tables

**Figure 1 jcm-11-03286-f001:**
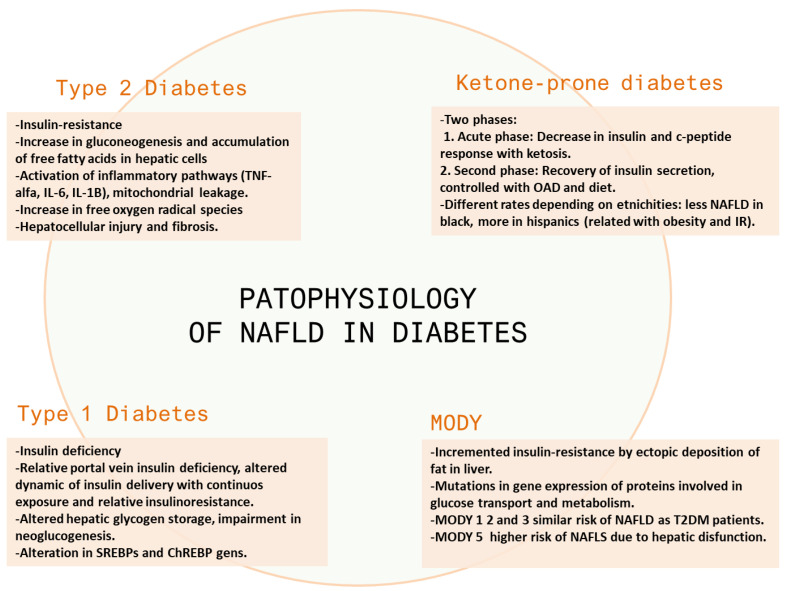
Main mechanisms involved in the physiopathology of NAFLD in different types of diabetes. OAD: oral antidiabetic drugs.

**Figure 2 jcm-11-03286-f002:**
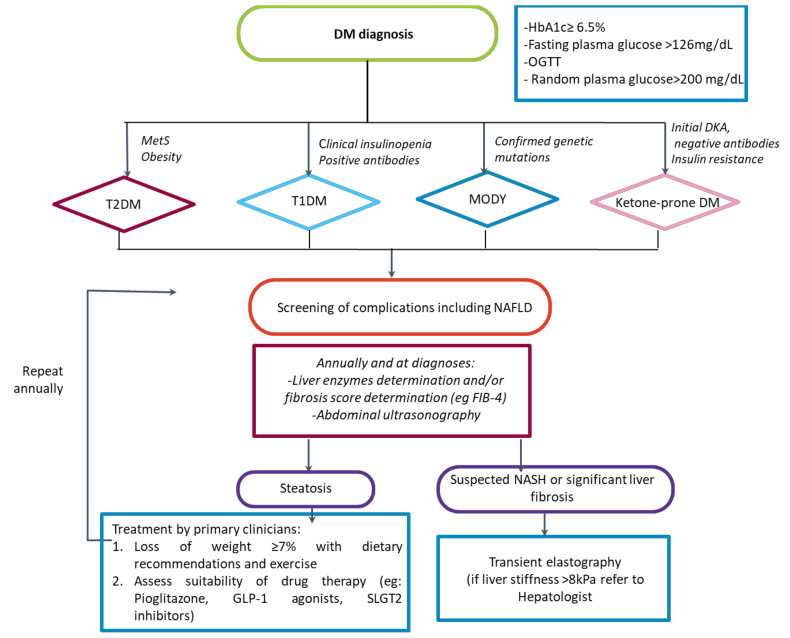
Algorithm proposed for the management of NAFLD in different typed of diabetes. DM: Diabetes mellitus; HbA1c: glycated hemoglobin; OGTT: Oral glucose tolerance test; MetS: Metabolic syndrome; T2DM: Type 2 diabetes mellitus; T1DM: Type 1 diabetes mellitus; MODY: Maturity-onset diabetes of the young; DKA: Diabetic ketoacidosis; OADs: Oral antidiabetic drugs; NAFLD: Non-alcoholic fatty liver disease; GLP1a: GLP-1 receptor agonists.

## Data Availability

Not applicable.

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
