# Peer review of "When Sugar Reaches the Liver: Phenotypes of Patients with Diabetes and NAFLD"

_jcm, 2022, doi:10.3390/jcm11123286_

Round 1

Reviewer 1 Report

This review covers a broad but very hot topic. The authors describe the pathophysiology and management of NAFLD in various diabetes phenotypes.

This reviewer raises some issues that should be addressed.

1- It is well known that IR is the strongest pathophysiological link between NAFLD, type 2 diabetes and cardiovascular disease. Very recent studies have shown that the reduction of IR through the sustained clearance of HCV by direct-acting antivirals leads to both a reduction in the onset of type 2 diabetes (Diabetes, Obesity and Metabolism, 2020, 22(12):2408–2416. doi: 10.1111/dom.14168) and clinical expressions of atherosclerosis (Atherosclerosis, 2020, 296:40–47. doi: 10.1016/j.atherosclerosis.2020.01.010 - Nutrition, Metabolism & Cardiovascular Diseases (2021) 31, 2345e2353. doi: 10.1016/j.numecd.2021.04.016). These interesting issues as well as the above references should be commented in discussion.

2- Intriguingly, recent biopsy studies have shown that steatohepatitis is the only feature of liver damage in type 2 diabetes (PLoS One. 2017 Jun 1;12(6):e0178473. doi: 10.1371/journal.pone.0178473). This important observation limits the ability to diagnose NASH with non-biopsy indices. This important issue should be commented on in the text.

3- At least one figure, perhaps even more, describing the pathophysiological relationship between the different phenotypes of diabetes and NAFLD would be appropriate in this review.

4- A linguistic revision by a native English speaker is required.

Author Response

First of all, we would like to thank you for your time and valuable suggestions that have considerably increased the value of the paper. Please find our answers below your comments.

Comment 1- It is well known that IR is the strongest pathophysiological link between NAFLD, type 2 diabetes and cardiovascular disease. Very recent studies have shown that the reduction of IR through the sustained clearance of HCV by direct-acting antivirals leads to both a reduction in the onset of type 2 diabetes (Diabetes, Obesity and Metabolism, 2020, 22(12):2408–2416. doi: 10.1111/dom.14168) and clinical expressions of atherosclerosis (Atherosclerosis, 2020, 296:40–47. doi: 10.1016/j.atherosclerosis.2020.01.010 - Nutrition, Metabolism & Cardiovascular Diseases (2021) 31, 2345e2353. doi: 10.1016/j.numecd.2021.04.016). These interesting Comments as well as the above references should be commented in discussion.

Answer: Thank you for your suggestion, we have added information about the link between HCV clearance and the reduction of the cardiovascular risk factors mentioned

Comment 2- Intriguingly, recent biopsy studies have shown that steatohepatitis is the only feature of liver damage in type 2 diabetes (PLoS One. 2017 Jun 1;12(6):e0178473. doi: 10.1371/journal.pone.0178473). This important observation limits the ability to diagnose NASH with non-biopsy indices. This important Comment should be commented on in the text.

Answer: Thank you very much for your suggestion. As recommended, we added the importance of biopsy in NASH diagnoses and some supporting data.

Comment 3- At least one figure, perhaps even more, describing the pathophysiological relationship between the different phenotypes of diabetes and NAFLD would be appropriate in this review.

Answer: As recommended, we added a resume figure of pathophysiological findings related with this topic.

Comment 4- A linguistic revision by a native English speaker is required.

Answer: Thank you very much. A native English speaker reviewed the paper.

Yours sincerely

Andreea Ciudin and Juna M Pericas

Reviewer 2 Report

This review describes the prevalence, pathophysiology, and clinical status of various types of diabetes mellitus and their association with NAFLD. This article is full of content and logically complete, but some descriptions are not very detailed. I still have a few concerns that need attention.

1. Please describe the full name of MODY diabetes in abstract when it first appears.

2. Several epidemiologic studies has demonstrated the relevance between T2DM with NAFLD, so please provide additional explanation for this part.

3. Besides insulin resistance, T2DM pathophysiology includes hyperglycemia. What impact does it have on NAFLD progression?

4. There are many antihyperglycaemic drugs for T2DM, such as Metformin, SGLT2 inhibitor PPAR agonists and so on. Would this kind of drugs have functions on NAFLD?

5. Lipodystrophy patients often have insulin resistance. Are they at high risk of NAFLD? Is the mechanism similar with T2D patients?

Author Response

First of all, we would like to thank you for your time and valuable suggestions that have considerably increased the value of the paper.

Comment 1: Please describe the full name of MODY diabetes in abstract when it first appears.

Answer: Data was added.

Comment 2: Several epidemiologic studies has demonstrated the relevance between T2DM with NAFLD, so please provide additional explanation for this part.

Answer: Thank you for your suggestion, we have included additional data obtained by cohort and prospective studies of the epidemiology between NAFLD and T2DM.

Comment 3: Besides insulin resistance, T2DM pathophysiology includes hyperglycemia. What impact does it have on NAFLD progression?

Answer:  According to your suggestion we added some additional information about the role of hyperglycaemia in inflammation and hepatic fibrosis.

Comment 4: There are many antihyperglycaemic drugs for T2DM, such as Metformin, SGLT2 inhibitor PPAR agonists and so on. Would this kind of drugs have functions on NAFLD?

Answer:  Thank you for your comment, we included more information about the different pharmacological therapies used in NAFLD treatment RCT.

Comment 5: Lipodystrophy patients often have insulin resistance. Are they at high risk of NAFLD? Is the mechanism similar with T2D patients?

Answer: According to your suggestions, because of the close link between lipodystrophy and insulin resistance, we added more information about this topic.

Your sincerely, 

Andreea Ciudin and JuanM Pericas

Reviewer 3 Report

The topic is interesting and the authors correctly described an information of DM and NAFLD.

The main weak point is the lack of methodological section. According to „Instructions for Authors” systematic reviews published in JCM should follow the PRISMA guidelines. I suggest that the authors should add at least informations such as: inclusion and exclusion criteria for the review, databases, registers, websites, organisations, reference lists and other sources searched or consulted to identify studies, methods used to collect data from reports, and others.

In 2.1 section „Type 2 diabetes and obesity – the metabolic syndrome paradigm”, authors can emphasise the role of proper diet in the treatment of T2DM. Information about and improving clinical indicators after changing lifestyle also should be added.

Line 29: MODY should be explained

Line 76: „Type 2 diabetes is the most common form of T2DMshouldn't be DM?

Line 93, 94: „kg/m2 „ 2 should be in a square

Author Response

First of all, we would like to thank you for your time and valuable suggestions. We have included a methodological section, that considerably increased the value of the paper.

Comment 1: In 2.1 section „Type 2 diabetes and obesity – the metabolic syndrome paradigm”, authors can emphasise the role of proper diet in the treatment of T2DM. Information about and improving clinical indicators after changing lifestyle also should be added.

Answer: Thank you for your comment, we included more information about the data obtained in different prospective studies about lifestyle changes and T2DM and NAFLD development. 

Comment 2: Line 29: MODY should be explained.

Answer: Changed accordingly. Thank you.

Comment 3: Line 76: „Type 2 diabetes is the most common form of T2DM” shouldn't be DM?

Answer: We have changed accordingly.

Comment 4: Line 93, 94: „kg/m2 „ 2 should be in a square.

Answer:  We have changed accordingly.

Your sincerely,

Andreea Ciudin and Juan M Pericas

Round 2

Reviewer 1 Report

No further comments.